# Comparison of virtual reality and real environment effects on perception of height in healthy individuals

Eda Karaman[1]*, Oğuz Yılmaz[2], Serkan Eti[3]

1 Department of Audiology, Graduate School of Health Sciences, Istanbul Medipol University, Istanbul, Türkiye, 2 Department of Audiology, Faculty of Health Sciences, Izmir Bakırçay University, Izmir, Türkiye, 3 Department of Computer Programming, School of Vocational Studies, Istanbul Medipol University, Istanbul, Türkiye

* ecevik@medipol.edu.tr

## Abstract

### Purpose

The aim of this study was to evaluate the effects of mechanically stimulated sacculus on our height perception.

### Methods

Between 1.09.2022 and 30.06.2023, 52 volunteers, 27 women and 25 men, aged 20–50 years, were included in the study. Pure tone audiometry test, acoustic immittance, vestibular evoked myogenic potentials (VEMP) and mini mental tests (MMSE) were performed on these individuals. Afterwards, height estimations were made by looking from top to bottom and from bottom to top using mechanical stimulation in real environment and elevator simulation in virtual reality (VR) environment. Participants were informed in writing with an informed consent form and their signed consent was obtained.

### Results

The averages of the height estimates made in the VR environment and in the real environment were compared with each other and no significant difference was observed ($p > 0.05$). When the height estimations made in the VR environment and in the real environment were compared with the current height value, a significant difference was observed only in the height estimation made by looking from the bottom up in the VR environment, and it was found to be higher than the current height ($p < 0.05$). When the height estimation values in the VR environment and in the real environment were compared with the place where height estimation was started, no significant difference was observed ($p > 0.05$).

**Data availability statement:** All relevant data for this study are publicly available from the Zenodo repository (https://doi.org/10.5281/zenodo.16754317).

**Funding:** External funding for this study was provided by Istanbul Medipol University Scientific Research Projects Support Program with project number 2022/37.

**Competing interests:** The authors have declared that no competing interests exist.

## Conclusion

In our study, the effect of the mechanical effect of the saccule on the height perception was investigated, and no significant difference was obtained in the height estimates made in the VR environment and in the real environment. Mechanical stimulation of the saccule is thought to have a limited role in height perception.

## Introduction

Balance is defined as the ability to control the centre of gravity on the support surface [1]. It is also responsible for providing spatial orientation to determine one's position and orientation in the world [2].

Cognitive skills basically depend on the perception of the external world and the formation of reasoning abilities based on these perceptions. Depending on the external world and the processing of the received data, we perceive our environment with egocentric and exocentric approaches and make judgments accordingly [3]. In this context, egocentric mechanisms are used to process information about the state of objects in the environment relative to the person, while exocentric mechanisms are used to evaluate the position of objects in the environment relative to each other and to the person. The disruption of sensory perceptions or the problem in their processing disrupts our perception of the environment [4].

One of the senses we use to perceive the environment is the vestibular system. Studies have shown that disorders in the vestibular system reduce cognitive skills. Among the vestibular system structures, especially otolith organ disorders are observed to be prominent in this field [5].

Otolith organs consist of cell structures in the inner ear that basically respond to linear accelerations of the hairy cells in its structure. One of these otolith organs, the saccule, causes reactions to increased stimulation, especially as a result of mechanical bending of hairy cells during up and down movements of individuals [5]. A test battery called cervical vestibular evoked myogenic potentials (cVEMP) is used to evaluate the saccule.

VR is frequently used in many fields today, including healthcare. In the VR system, it creates an environment where people's visual perceptions are altered. VR systems simulate these movements in the worlds they create and provide the perception of the environment with visual and audio stimuli depending on the current technical inadequacies. The visual system stands out as the most important data collection system in this field. In terms of ensuring visual constancy, which is one of the main tasks of the vestibular system, it can be considered that the angular axial movements of the head, especially in the semicircular canals, can contribute to VR environments in this field. Despite this, the mechanical effects of the otolith organs, which especially affect cognitive functions, are simulated with VR systems [6]. This issue raises the question of how successful the increasingly widespread simulation programs (flight simulations, surgical simulations, etc.) are in the perception of our environment and the effects of perceptual changes in VR environments. Previous studies have

demonstrated the effects of visual perception changes on saccule functions [7,8], but the effects on height perception have not been adequately evaluated.

It has been reported that high-level cognitive functions (such as spatial perception) are involved in the formation of reflex responses in the balance system [9]. Studies have shown that there is a connection between the balance system and cognitive skills [10]. In this context, it is stated that the balance system can be affected by the person's age, illnesses, etc. In addition to peripheral pathologies, central pathologies and neurodegenerative diseases are also thought to cause balance problems [11]. Studies have shown that VEMP results are abnormal when compared with healthy individuals [12–14]. In neurodegenerative diseases, it has been shown that the balance system and visuospatial perception are affected and therefore, an increase in inaccuracy in distance estimation may be observed in these patients.

Although the interactions between cognitive skills and VEMP studies are known, there is no study investigating the functions of vestibular structures in perception, which is a part of the cognitive process. There is only one study on height perception, and in this study, people with fear of heights were asked to estimate the height and size of an object by looking from a balcony [15]. Studies on height perception indicate that this perception changes especially in people with fear of heights. However, there are no studies on the changes in height perception in cases where the saccule is damaged or the saccule is not used. The fact that the saccule is not mechanically stimulated in VR environments suggests that it may cause changes in height perception. The aim of this study is to evaluate the effects of VR applications on height perception in individuals.

## Materials and methods

This study is an experimental research and a randomized controlled trial. This study was conducted at the South and North Campus of Istanbul Medipol University (IMU) and participants were recruited between 1.09.2022 and 30.06.2023. At the same time, ethical approval for this study was obtained from the IMU Non-Interventional Clinical Research Ethics Committee on 30.09.2021 with decision number 964. The sample size was determined by g-power analysis and 50 people were found. In this context, a total of 52 people, 27 women and 25 men, were included in the study. This study included participants between the ages of 20 and 50 with normal hearing, as confirmed by pure tone audiometry, Type A tympanograms, and acoustic reflexes. Participants also exhibited positive cVEMP results, no defined balance or visual impairment, and no fear of heights. Participants who did not respond to the cVEMP test and who scored below 24 points in the MMSE were excluded from the study. Participants were informed in writing with an informed consent form and their signed consent was obtained.

VR environment creation: The simulation, which was shown to the participants in a VR environment, was specially prepared for this study. A short preliminary study was carried out while selecting the application area. In this study, it was primarily observed that people's use of exocentric mechanisms in height perception was at the forefront. Generally, when individuals were estimating their height, they calculated how high they were by multiplying the number of floors they were on by the approximate floor height. In order to avoid the use of this mechanism, we have chosen a building area where the floors cannot be calculated by looking up or down (Fig 1, Fig 2). In addition, an environment was created in which there were no floor numbers in the VR elevator so that participants could not determine which floor they would be going to. In order to avoid a similar effect, the participants were not informed which floor they were on during the test, including in the elevator. In the preparation phase, the data to be sent to the designer was first collected. The ground clearance of the 5th floor was measured using a Bosch plr 25 laser measure (Germany) and found to be 24 metres. The plan of the 5th floor, the exterior of the school and the videos of the objects inside the elevator were taken; at the same time, the time spent by the elevator from the ground floor to the 5th floor was calculated and sent to the designer to be made in accordance with the reality. The school environment was exactly reflected in the simulation and the time the participant spent in the elevator while going up to the 5th floor, the height of the 5th floor, and the distance s/he travelled to the window after getting off the elevator were prepared in a way to be compatible with the real environment and not to provide a clue to the

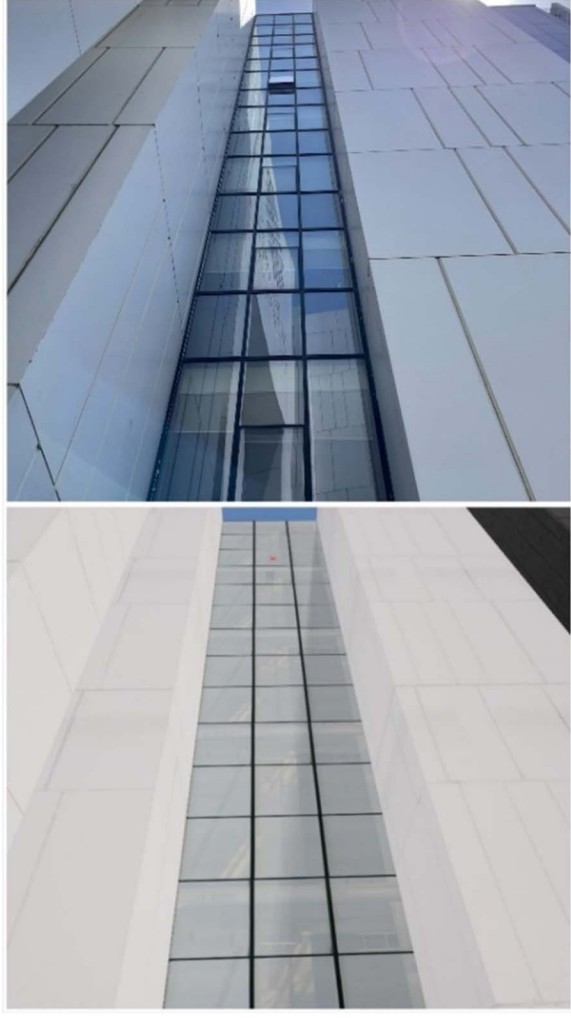

**Fig 1. Bottom-up view in real and virtual reality environments.**

participant to understand which floor s/he was on. The prepared simulation was shown to the participants using Oculus Quest 2 (USA) glasses.

Preparation for the experiment: Pure tone audiometry (Interacoustics, AC40 model (Denmark)), tympanometry test (Interacoustics, Titan device (Denmark)), cVEMP test (Intelligent Hearing Systems/IHS (Miami, FL, USA) Smart EP 5.10) were performed in all participants. At the same time, a MMSE was applied to test cognitive skills. The height of the participants was measured with a tape measure at the highest point of the head in an upright position against a wall.

Applied tests: The cVEMP and oVEMP tests were applied to the participants, and utricle and saccule functions were evaluated. Before electrode placement, the skin was cleaned with NuPrep gel, the active electrode was placed on the forehead, the reference electrodes were placed on the middle 1/3 of the sternocleidomastoid (SCM) muscles and the ground electrode was placed on the sternum. The electrode impedance was controlled to be in the range of 0–5 kΩ. Bilateral 'E-A-RTONE™ 3A Insert Earphone' was used for auditory stimulus transmission. In a seated position on the stretcher, participants turned their heads in the opposite direction to the ear being tested to contract the SCM muscle. During the test, a 500 Hz Tone Burst stimulus was given at 100 dB nHL and a double trace with 200 sweeps was recorded.

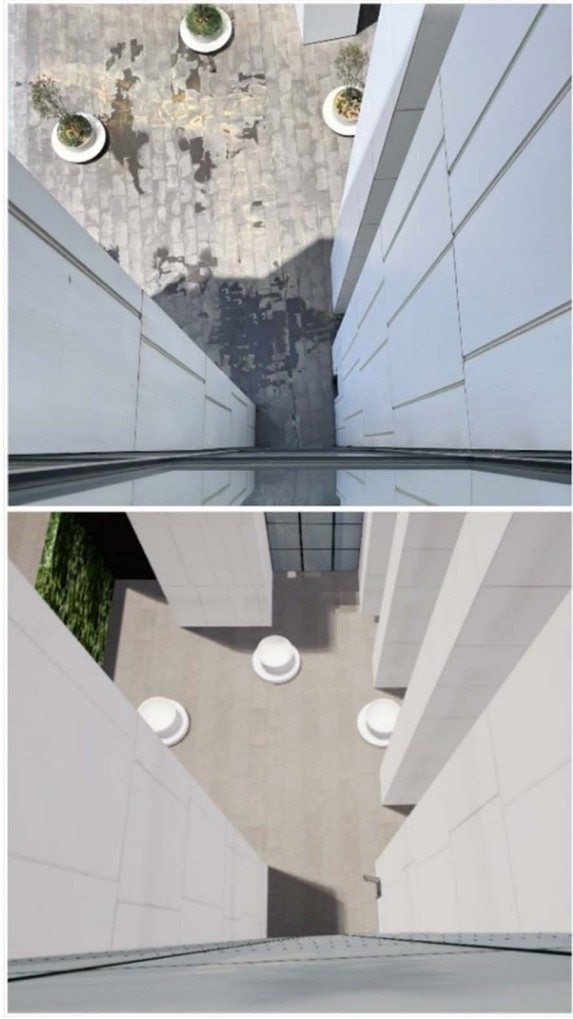

**Fig 2. Top-down view in real and virtual reality environments.**

The MMSE is a widely used scale for the assessment of cognitive skills. Turkish validity and reliability study was conducted by Güngen et al. Participants were informed, and personal information was obtained before the test application. Tasks in the categories of orientation, recording memory, attention and calculation, recall and language were carried out in accordance with the relevant guideline. Participants with a total score between 24–30 as a result of the test were included in the study. During the implementation, attention was paid to the instructions and time limits.

Experiment implementation: During the experiment, participants were asked to estimate height. This estimation was designed to take place when looking down from above and when looking up from below. Here, in order to prevent the aforementioned exocentric mechanisms from being activated, care was taken to ensure that there was no floor number in the elevator and that they could not calculate the number of floors as mentioned in the VR environment. In the real environment, participants were blindfolded, and their eyes were not opened until they reached the edge of the window to avoid seeing floor numbers in the corridors, including the elevator. In addition to this, to eliminate the learning effect, half of the participants were randomly asked to make height estimations by first looking up from the bottom at preset heights and then looking down from the top by going up with the help of an elevator. The other half of the participants were first taken

up by elevator and asked to estimate the height by looking from top to bottom. They were then taken down by elevator and asked to estimate the height by looking from bottom to top. This procedure was repeated in the VR and real environment conditions. (Fig 1, Fig 2)

Statistical analysis: Statistical Package for Social Sciences (SPSS) Vs 25 (SPSS inc., Chicago, IL, USA) was used for statistical analysis. "Kolmogorov-Smirnov Test" was used to evaluate the conformity of the data to normal distribution. The height estimates made by using the real elevator and by looking down and up in the VR environment were compared separately for both environments and the "Wilcoxon Sign Rank Test" was used. The height estimates made in the VR environment and in the real environment were compared with the place where the height estimation was started using the "Mann-Whitney U test". A value of $p < 0.05$ was accepted for statistical significance. The proposed sample size is 48 participants, which holds a power of 80%, significance level of $\alpha = 0.05$ and detect minimum clinically important difference.

## Results

When the values obtained from the height estimations made in the VR environment and in the real environment were compared, no statistically significant difference was obtained between the averages of looking up from below in the real and VR environment and looking down from top in the real and VR environment ($p > 0.05$). (Table 1) (Fig 3).

The data obtained from the height estimates made in the VR environment and in the real environment were compared with the current height value using "Wilcoxon". Since the height of the individuals' gaze also affects the estimation, their height from eye level was taken into account. Comparing the average values of the height estimation values of the participants by looking down from top in the VR environment and in the real environment with the current height + height of the person, no statistically significant difference was obtained ($p > 0.05$). There was no statistically significant difference

**Table 1. Comparison of the values of height estimations obtained from bottom-up and top-down view in real environment and virtual reality environment.**

|  | Mean | Sd. | z | p value |
|---|---|---|---|---|
| R-bottom-up view | 26.94 | 12.837 | −0.152 | 0.879 |
| VR-bottom-up view | 28.721 | 15.882 |  |  |
| R-Top-down view | 28.65 | 16.302 | −0.876 | 0.381 |
| VR-top-down view | 30.529 | 16.871 |  |  |

R: Real Environment, Sd: Standard Deviation, VR: Virtual Reality

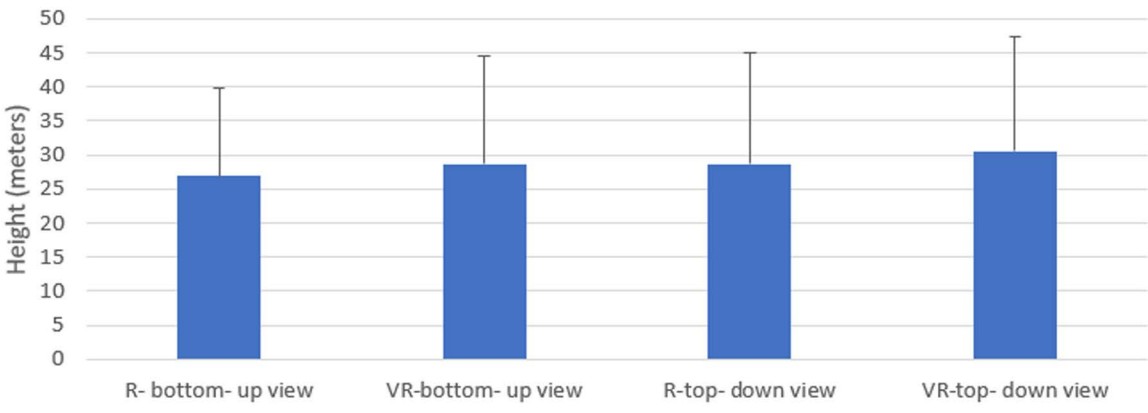

**Fig 3. Comparison of the values of the height estimates obtained in the real environment and in the virtual reality environment by looking from bottom to top and top to bottom.**

(p > 0.05) when the average values of the height estimation values made by looking up from the bottom in the real environment were compared with the current height – the height of the person. A statistically significant difference was obtained when the average of the height estimation values made by looking from bottom to top in the VR environment was compared with the current height – the height of the person (p < 0.05). When the difference between the data was analyzed, the average of the height estimation made by looking from bottom to top in the VR environment (mean. 28.721, sd. 15.882) was found to be significantly higher than the current height – height of the person (mean. 22.302, sd. 0.0882) (Table 2) (Fig 4).

There was no statistically significant difference (p > 0.05) when the data obtained from the height estimations made in the VR environment and in the real environment by looking up from the bottom and down from the top were compared with the place where the height estimation was started (Table 3) (Fig 5).

## Discussion

The vestibular system provides information about linear acceleration with reference to the gravitational force with inputs from the utricle and saccule. It is known that the saccule is more sensitive to vertical accelerated movements [16]. In addition, otolith organs are also reported to have an effect on spatial memory and social cognition. Studies have shown that utricle and saccule contribute to spatial learning and memory [17–19]. VR is a computer-based technology that gives users a sense of reality in an interactive environment and is widely used in the field of health. At the same time, virtual environments are powerful tools that enable individuals to test their theories of perception and spatial cognition [20,21].

In a study conducted by Öztürk et al., VEMP and Video Head Impulse Test (VHIT) tests were applied to the participants by creating a situation in which visual perception was altered using optical illusion and visual and vestibular data contradicted each other. When the test results performed in the presence and absence of optical illusion were compared, changes in VEMP responses were observed [7]. In another study, the cVEMP test was applied to the participants before, during and after accelerated movement in the vertical plane in a VR environment. This vertical movement in a VR environment affected the responses of the saccule and increased the amplitude of cVEMP responses [8]. The aforementioned studies have shown that saccule functions are affected by changes in visual perception. However, the underlying cause of

Table 2. Comparison of height estimations made by looking up from bottom and down from top in real and virtual reality environments with the current height value.

| | Mean. | Sd. | z | p value |
|---|---|---|---|---|
| Current Height – Participant's Height | 22.302 | 0.088 | −2.841 | **0.004*** |
| VR-Bottom-up view | 28.721 | 15.882 | | |
| Current Height – Participant's Height | 22.302 | 0.088 | −1.912 | 0.056 |
| R-bottom-up view | 26.94 | 12.837 | | |
| Current height + Participant's Height | 25.697 | 0.088 | −1.202 | 0.229 |
| VR-Top-down view | 30.529 | 16.871 | | |
| Current Height + Participant's Height | 25.697 | 0.088 | −0.009 | 0.993 |
| R-top-down View | 28.65 | 16.302 | | |

R: Real Environment, Sd: Standard Deviation, VR: Virtual Reality

*p < 0,005

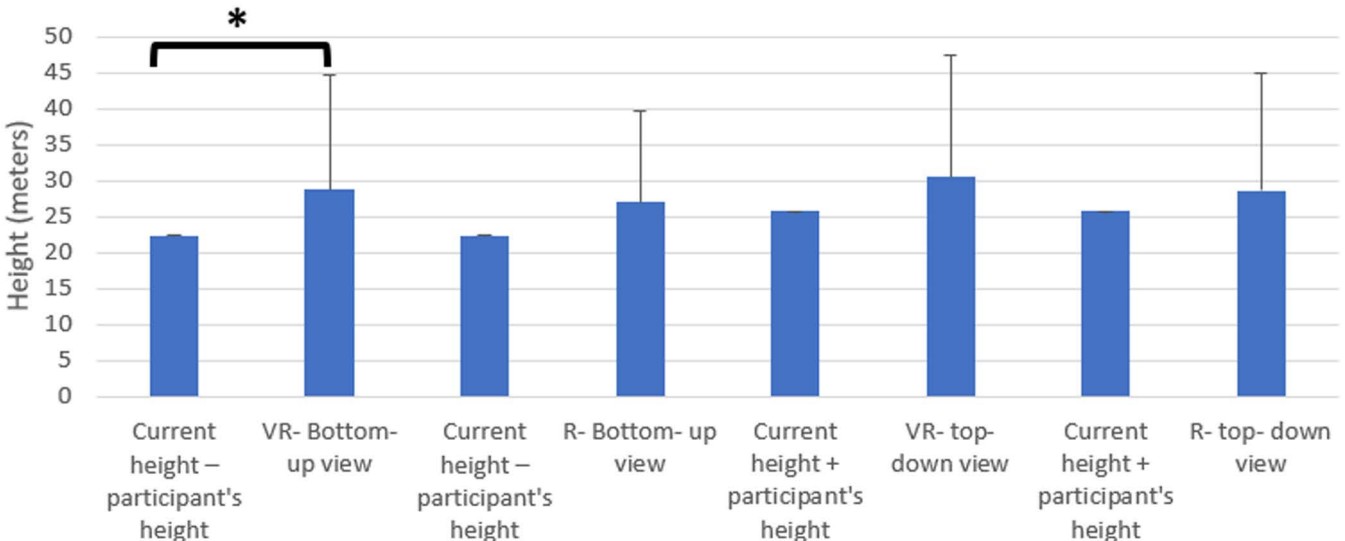

**Fig 4. Comparison of the height estimates made by looking from bottom up and top down in real and virtual reality environments with the current height value.**

**Table 3. Mann-Whitney U test results for the comparison of the height estimation values in the virtual reality environment and in the real environment with the place where the height estimation started.**

|  | Starting Point | n | Mean | Sd. | U | z | p value |
|---|---|---|---|---|---|---|---|
| VR- bottom-up view | Ground floor | 26 | 27.673 | 15.796 | 327.000 | −0.202 | 0.840 |
|  | 5th floor | 26 | 29.769 | 16.212 |  |  |  |
| VR-top-down view | Ground floor | 26 | 32.346 | 17.274 | 287.500 | −0.926 | 0.354 |
|  | 5th floor | 26 | 28.712 | 16.595 |  |  |  |
| R-bottom-up view | Ground floor | 26 | 29.269 | 15.966 | 293.000 | −0.825 | 0.409 |
|  | 5th floor | 26 | 24.615 | 8.367 |  |  |  |
| R-top-down view | Ground floor | 26 | 33.577 | 20.227 | 261.500 | −1.405 | 0.160 |
|  | 5th floor | 26 | 23.731 | 9.089 |  |  |  |

R: Real Environment, Sd: Standard Deviation, VR: Virtual Reality

these effects is unknown. In our study, unlike these studies, we created an environment that alters visual perception but does not mechanically stimulate the saccule and evaluated its effect on height perception. Although we were expecting a better estimation of the height with the mechanical stimulation of the saccule, we found no changes. When evaluated together with previous studies, this suggests that the mechanical influence of the saccule is important in the perception of elevation, whereas visual influences are more important than mechanical stimuli in the perception of height.

In a meta-analysis study conducted to investigate the effect of the screen worn on the head of people in the perception of distance in VR, studies on the technical features of VR glasses were examined in general. As a result of 131 studies analyzed, it was reported that inadequate perception persists even in currently sold VR glasses, but wider field of view, lighter weight and higher resolution have an effect on accurate distance perception [22]. Dimensions and distances in virtual environments were perceived less than in the real environment. This difference is further reduced when newer equipment is used, but the underlying perceptual reasons for this are not fully understood [23]. In our study, vertical distance estimates made in the virtual environment were found to be higher than

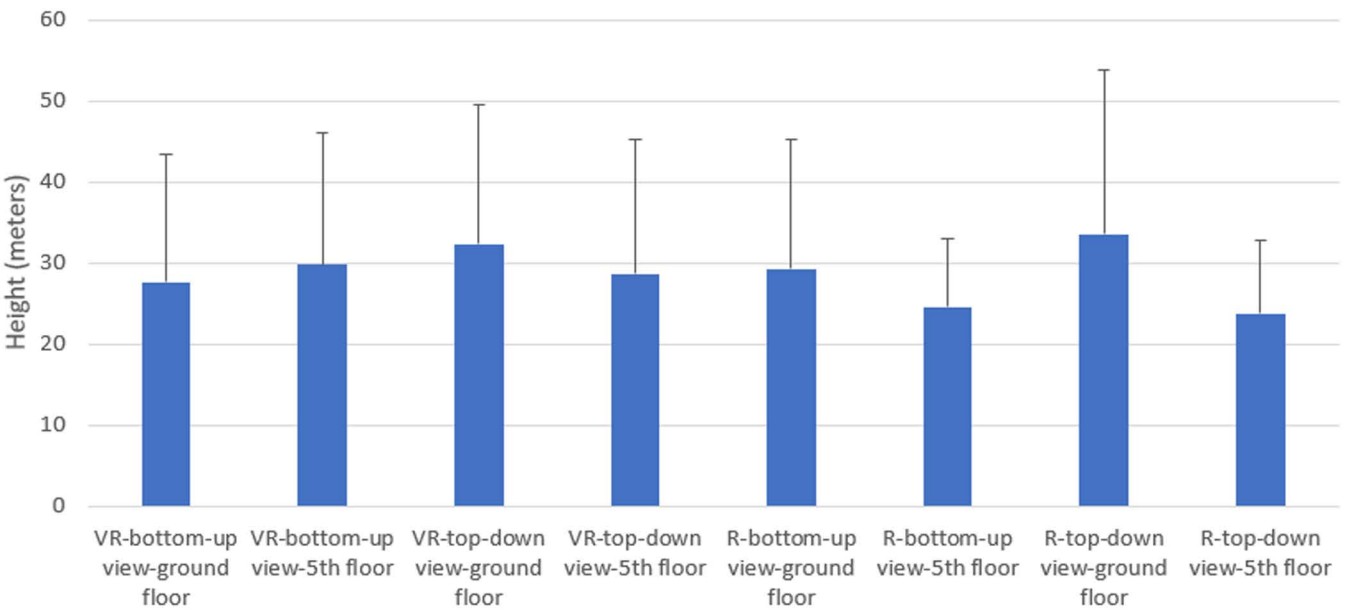

**Fig 5. Comparison of the height estimation values in the virtual reality environment and in the real environment with the place where height estimation started.**

those made in the real environment. It is thought that this may be due to the technological features of the equipment we used. In the meta-analysis study [22], it was stated that according to the glasses brands used, Oculus Rift DK1 produced 96% and 100% accurate distance perceptions, respectively, according to two studies using it [24,25]. In two studies using Pimax 5k Plus, it was reported that it produced 86% and 100% accurate distance perceptions [26,27] In studies using Oculus Rift DK2 [28], Oculus Rift CV1 [29] and HTC Vive [30], there are reports of accurate distance perceptions, but the averages of these brands are lower compared to the averages reported in the meta-analysis of all studies. We used Oculus Quest 2 VR glasses in our study. Compared to Oculus Rift DK1, Oculus Quest 2 is known to have better features such as resolution, viewing angle, wireless usage. Therefore, in our study, Oculus Quest 2 was thought to support the correct understanding of distance perception. With the developing technologies, these perceptions can be perceived more accurately in VR environment. At the same time, it is thought that the use of advanced technologies in applications such as vestibular rehabilitation in studies such as games etc. may be beneficial.

In our study, it was concluded that saccule did not affect the perception of height, but further research is needed in people with neurodegenerative diseases. Since there is no change in perception in VR environment, we think that the use of VR environment instead of real environment for improving perception in neurodegenerative diseases and patients with perceptual disorders will not cause a problem. Our study is the first study to examine the possible cause of perception changes. It is thought that it may be possible to evaluate how these processes change with studies to be conducted on people with neurodegenerative diseases in the future.

## Conclusion

In our study, we think that the role of mechanical stimulation of the saccule in height perception is limited. However, it is thought that it may be important in terms of developing a rehabilitation technique with the use of VR in neurodegenerative diseases and patients with perceptual disorders.

## Acknowledgments

We would like to thank participants for their patience and invaluable contribution to this study.

This article was produced from the thesis titled "Developing the Reality-Based Test Battery for Evaluating the Role of the Saccule in the Perception of Height".

## Author contributions

**Data curation:** Serkan Eti.

**Formal analysis:** Serkan Eti.

**Funding acquisition:** Oğuz Yılmaz.

**Methodology:** Eda Karaman, Oğuz Yılmaz.

**Project administration:** Oğuz Yılmaz.

**Validation:** Oğuz Yılmaz.

**Visualization:** Eda Karaman.

**Writing – original draft:** Eda Karaman.

**Writing – review & editing:** Eda Karaman, Oğuz Yılmaz.

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
