## [Decision Letter · Decision Letter 0]

29 Nov 2024

PONE-D-24-40618The role of saccule in height perceptionPLOS ONE

Dear Dr. Karaman,

Thank you for submitting your manuscript to PLOS ONE. After careful consideration, we feel that it has merit but does not fully meet PLOS ONE’s publication criteria as it currently stands. Therefore, we invite you to submit a revised version of the manuscript that addresses the points raised during the review process.

**Please see comments. **

We look forward to receiving your revised manuscript.

Kind regards,

Jeyasakthy Saniasiaya, MD, MMed ORLHNS, FEBORLHNS

Academic Editor

PLOS ONE

Journal Requirements:

“External funding for this study was provided by Istanbul Medipol University Scientific Research Projects Support Program with project number 2022/37.”

4. In this instance it seems there may be acceptable restrictions in place that prevent the public sharing of your minimal data. However, in line with our goal of ensuring long-term data availability to all interested researchers, PLOS’ Data Policy states that authors cannot be the sole named individuals responsible for ensuring data access (http://journals.plos.org/plosone/s/data-availability#loc-acceptable-data-sharing-methods).

6. Please ensure that you refer to Figure 5 in your text as, if accepted, production will need this reference to link the reader to the figure.

**Additional Editor Comments:**

MAjor grammatical and language editing required

Hypothesis of this study needs to be justified

Methodology shoudl be more detailed

Reviewers' comments:

Reviewer's Responses to Questions

**Comments to the Author**

1. Is the manuscript technically sound, and do the data support the conclusions?

Reviewer #1: Yes

Reviewer #2: No

2. Has the statistical analysis been performed appropriately and rigorously? 

Reviewer #1: Yes

Reviewer #2: I Don't Know

3. Have the authors made all data underlying the findings in their manuscript fully available?

Reviewer #1: Yes

Reviewer #2: Yes

4. Is the manuscript presented in an intelligible fashion and written in standard English?

Reviewer #1: Yes

Reviewer #2: No

5. Review Comments to the Author

Reviewer #1: Title: Consider changing title of the paper – as the given title is not adhere with the aim of the study.

‘Comparison of virtual reality and real environment effects on perception of height in healthy individuals’

Page 4 Line 78: what is meant by high perception??

Page 5 Line 104-106: Rephrase the sentence for easy understanding

Page 5: Give a sub heading for designing the simulation of virtual reality environment

Page 6: Explain the audiological procedures, cVEMP and MMSE test procedures in detail

Page 6: Give a sub heading for VR simulation environment and real environment

Page 7 Line 146: the point at which people started to estimate the height was also changed – how it is changed, explain this in detail.

Page 8 Line 177-180:

What is meant by current height? What is the defined value for the current height and how you defined it???

What is the height of the individuals who had participated in this study, and how it influences the height perception; Provide the appropriate information in discussion section with references. Also, every individual’s height will be different and how it is analysed statistically.

Table 3:

why only 26 participants were included, but in methods section it is mentioned that the study included 52 participants.

Why Mean and SD values mentioned with comma???

Page 12 Line 223: evaluated the effect of this, ‘this’ means???

Figure 3-5: y-axis of the bar graphs denotes???

Figure 5: why y axis values are in 5 digits???

General: Maintain the uniformity in mentioning the term ‘virtual reality’ or simply use the abbreviated term ‘VR’

General: Provide the expansion for the abbreviated terms used in the document for the first time.

Reviewer #2: This paper claims to look at the effect of mechanical stimulation of the saccule on height perception. However it is not very clear what is meany by mechanical stimulation. It seems to just mean going up in a lift, but the rationale as to why that should affect height perception is not given. Clearly the mechanical stimulation is not simultaneous with the height measurement, which is a limitation of the approach.

It is not clear to me that people would in general be reliable at height perception. That seems to be quite key to the experiment. There needs to be some evidence given that people are reliable at height perception, as if they are not, then the experiment is not likely to find anything significant.

I would like to see data as to how reliable people are at height perception – ideally for some different heights. The choice of the 5th floor seems fairly arbitrary. Presumably at some point (say 100th floor), perception of height becomes very unreliable, but it is not clear at what point that would be the case, or whether we expect height perception at the 5th floor to be reliable.

The background is generally not well written. Many sentences need work and do not entirely make sense. In general the English in the paper is not very good, which is somewhat frustrating when trying to review the work. I don’t think it is my role as a reviewer to correct English. The manuscript should be better proof-read before submission to a journal.

The hypothesis of the experiment is not well defined. Are you assuming that the lift is stimulating the saccule and that this will somehow affect height perception? Why? Have others shown that (and if so what did they do in their experiments?). It seems quite tenuous.

The points about correcting for the height of the subjects are not explained clearly/justified. There is no hypothesis stated, or any rationale given for the extra analysis.

I am not convinced that going 5 floors in a lift is stimulating the saccule to a degree that we would expect a change in height perception (if we are assuming that there is evidence for saccular stimulation affecting height perception – although that is not clear from the manuscript). Also the non-simultaneous measurement of height with the life stimulation means that any affect on height perception could have recovered anyway. I therefore don’t think that the conclusion ‘Mechanical stimulation of the saccule is thought to have a limited role in height perception’ is justified from the work carried out.’

Given these limitations I don’t think that the work is suitable for publication at this time. Possibly the work comparing VR height perception to real-world height perception would be of interest to some in the VR community.

Abstract

You probably don’t need to put the date.

It might help to say briefly what the mechanical stimulation is in the abstract.

Acoustic immittance (not ‘immitansmetry’)

L53 I am not sure about your definition of balance

L54 which image?

L64/65 I don’t understand why visual perception should alter saccular function. Can you explain more?

L69 can you be more specific what the link is between balance and cognitive skills

L69 poor sentence ‘it is stated that the balance system can be affected by the age of a person, his/her disorders, etc.’

L72 poor sentence ‘Studies have shown that VEMP results were the most effected test batteries by abnormal when compared with healthy individuals.’

L73-75 can you reference the evidence for this and explain conceptually why this should occur?

L76 what is the link between cognitive skills and VEMP? Can you explain this and reference it?

L78 High not heigh?

L82-84 Sentence needs work

Can you explain conceptually why the saccule should affect height perception?

106-108 Did you exclude subjects from the study?

128-131 I am not convinced that subjects would be accurate in height estimation from a 5th floor in general. Do you have any evidence that subjects are reliable in doing this?

P173 why is Wilcoxen in “”? Do you mean the Wilcoxen test?

P9 I don’t really understand the point of the analysis here. Won’t the VR and real world estimated include the height of the subjects?

I don’t really follow the point of Table 3 either. Why are you using a Mann-Whitney test here when you used Wilcoxen before.

6. PLOS authors have the option to publish the peer review history of their article (what does this mean?). If published, this will include your full peer review and any attached files.

Reviewer #1: No

Reviewer #2: No

---

## [Author Response · Author response to Decision Letter 1]

8 Feb 2025

Mrs. Jeyasakthy Saniasiaya,

Academic Editor

PLOS ONE

Dear Mrs. Jeyasakthy Saniasiaya,

According to your last email, please find attached a copy of the revised version of our paper, as a possible publication in PLOS ONE:

“THE ROLE OF SACCULE IN HEIGHT PERCEPTION” The comments and suggestions for revision are reflected in the current version of the paper.

A response letter with main changes has been done in this revision together a separated individual response for each reviewer to their comments and a specific reference section to support our answers are attached.

Looking forward to hearing from you.

Thank you.

Yours sincerely,

Response letter to the Reviewers’ Comments on PLOS ONE

Title of the paper: The Role of Saccule in Height Perception

First of all, authors would like to thank all the anonymous Reviewers and the Editors again for their efforts and valuable time to review and improve our paper.

Taking into account their constructive suggestions and comments, the paper has been carefully revised following the referees’ comments.

The main changes in the revised manuscript are:

1. Grammar and language organization

2. Introduction to the study

3. Materials and methods of the study

We provide below the responses to each referee’s comments.

Editor Comments

1. Major grammatical and language editing required

The manuscript edited by an native speaker after artificial intelligence check.

2. Hypothesis of this study needs to be justified

The introduction was rewritten to clarify the hypothesis and the objectives of the study.

3. Methodology should be more detailed

Edits have been made in the material and method section.

Reviewer #1

Reviewer’s Comment 1: Title: Consider changing title of the paper – as the given title is not adhere with the aim of the study.

‘Comparison of virtual reality and real environment effects on perception of height in healthy individuals’

The title of the article has been edited according to your suggestion. “Comparison of virtual reality and real environment effects on perception of height in healthy individuals’

Reviewer’s Comment 2: Page 4 Line 78: what is meant by high perception??

The stated section aims to define the perception of height.

In addition, in the aforementioned study, when estimating the size of objects, individuals may see objects smaller or larger than they actually are due to fear of height. Fear of heights can lead to distortion of individuals' perception when looking from a high place. This psychological effect also changes size estimations by affecting the perception of height.

Reviewer’s Comment 3: Page 5 Line 104-106: Rephrase the sentence for easy understanding.

In line with your suggestion, the specified part has been rearranged as “This study included participants between the ages of 20 and 50 with normal hearing, as confirmed by pure tone audiometry, Type A tympanograms, and acoustic reflexes. Participants also exhibited positive cVEMP results, no defined balance or visual impairment, and no fear of heights.”

Reviewer’s Comment 4: Page 5: Give a sub heading for designing the simulation of virtual reality environment.

In line with your suggestion, a sub-heading has been added to the material and method section. These sections were: a. VR environment creation, b. Preparation for the experiment:, c. Experiment implementation: d. Statistical analysis:

Reviewer’s Comment 5 Page 6: Explain the audiological procedures, cVEMP and MMSE test procedures in detail In general, the SOT test gives information about the data use of patients in pathologies. This is in line with our work objective. In addition, it was evaluated that the other tests mentioned would not be reliable due to the fatigue frequently observed in MS patients.

Test procedures were explained in detail in the test preparation section of the material and methods chapter.

Reviewer’s Comment 6: Page 6: Give a sub heading for VR simulation environment and real environment.

The criteria and specifications for the VR test environment are specified in the “VR environment creation” section.

Reviewer’s Comment 7: Page 7 Line 146: the point at which people started to estimate the height was also changed – how it is changed, explain this in detail..

All the procedure rephrased as “During the experiment, participants were asked to estimate height. This estimation was designed to take place when looking down from above and when looking up from below. Here, in order to prevent the aforementioned exocentric mechanisms from being activated, care was taken to ensure that there was no floor number in the elevator and that they could not calculate the number of floors as mentioned in the virtual reality environment. In the real environment, participants were blindfolded and their eyes were not opened until they reached the edge of the window to avoid seeing floor numbers in the corridors, including the elevator. In addition to this, to eliminate the learning effect, half of the participants were randomly asked to make height estimations by first looking up from the bottom at preset heights and then looking down from the top by going up with the help of an elevator. The other half of the participants were first taken up by elevator and asked to estimate the height by looking from top to bottom. They were then taken down by elevator and asked to estimate the height by looking from bottom to top. This procedure was repeated in the virtual reality and real environment conditions. (Figure 1, Figure 2)”

Reviewer’s Comment 8: Page 8 Line 177-180: What is meant by current height? What is the defined value for the current height and how you defined.

We want the define current height by 24 meter which is defined in materials and methods section as “The ground clearance of the 5th floor was measured using a Bosch plr 25 laser measure (Germany) and found to be 24 metres.”

Reviewer’s Comment 9: What is the height of the individuals who had participated in this study, and how it influences the height perception; Provide the appropriate information in discussion section with references. Also, every individual’s height will be different and how it is analysed statistically.

Within the scope of the study, the distances between the people and the place they look at change as height plus height subtracted from height due to the position of the eyes in the upward gaze and the distance looked at as height plus height due to the position of the eyes in the downward gaze. In order to eliminate this effect, the calculations given in Table 2 were activated to exclude the neck effect on distance changes

The height of the participants was measured with a tape measure at the highest point of the head in an upright position against a wall. This part added the material and methods section.

Reviewer’s Comment 10: Table 3: why only 26 participants were included, but in methods section it is mentioned that the study included 52 participants.

When estimating height in the study, half of the participants looked down from above and the other half primarily looked up from below. Therefore, Table 3 includes 26 participants whose starting point was the ground floor (look up from below) and 26 participants whose starting point was the 5th floor(look down from above), for a total of 52 participants.

Reviewer’s Comment 11: Why Mean and SD values mentioned with comma???

Page 12 Line 223: evaluated the effect of this, ‘this’ means???

It was found that the mean and standard deviation values were written with commas in some of the tables and with periods in others, and this inconsistency was corrected and all tables were arranged in accordance with the journal's spelling rules.

The section has rephrased as “The aforementioned studies have shown that saccule functions are affected by changes in visual perception. However, the underlying cause of these effects is unknown. In our study, unlike these studies, we created an environment that alters visual perception but does not mechanically stimulate the saccule and evaluated its effect on height perception.”

Reviewer’s Comment 12: Figure 3-5: y-axis of the bar graphs denotes???

Necessary adjustments were made to the figures to determine the quantities. In Figures 3, 4 and 5, the y-axes represent the elevation (in meters).

Reviewer’s Comment 13: Figure 5: why y axis values are in 5 digits???

Necessary adjustments were made to the figures to determine the quantities.

Reviewer’s Comment 14: General: Maintain the uniformity in mentioning the term ‘virtual reality’ or simply use the abbreviated term ‘VR’

Necessary arrangements have been made

Reviewer’s Comment 15: General: Provide the expansion for the abbreviated terms used in the document for the first time.

Necessary arrangements have been made

Reviewer #2

This paper claims to look at the effect of mechanical stimulation of the saccule on height perception. However it is not very clear what is meany by mechanical stimulation. It seems to just mean going up in a lift, but the rationale as to why that should affect height perception is not given. Clearly the mechanical stimulation is not simultaneous with the height measurement, which is a limitation of the approach.

Saccule stimulation is mainly based on the activation of hairy cells as a result of mechanical impact. Height perception is basically based on the opinions of individuals. However, it has been investigated whether the saccule, which plays a critical role especially in the perception of upward movements, has an additional effect on the perception of these movements. In this context, we focused on the question of whether the perception of upward movements produces a change in the perception of the person depending on the movement, other than just simultaneous perception. In this context, instead of simultaneous measurements, we aimed to evaluate the effect of fluid movement in hairy cells caused by a mechanical effect and the changes created by this movement in hairy cells on height perception.

It is not clear to me that people would in general be reliable at height perception. That seems to be quite key to the experiment. There needs to be some evidence given that people are reliable at height perception, as if they are not, then the experiment is not likely to find anything significant. I would like to see data as to how reliable people are at height perception – ideally for some different heights. The choice of the 5th floor seems fairly arbitrary. Presumably at some point (say 100th floor), perception of height becomes very unreliable, but it is not clear at what point that would be the case, or whether we expect height perception at the 5th floor to be reliable.

This study is based on the assumption that people's height perception is reliable. Although we could not find a direct study on this issue, observations made during the research suggest that participants' perceptions of height and distance are often based on similar strategies (e.g., estimation based on a known distance). It was observed that participants frequently used egocentric and exocentric evaluation mechanisms, especially when estimating the height of apartments.

In egocentric evaluations, participants made estimates with reference to their height (e.g., “If I am 1.75 m tall, that is about 4 times as tall, i.e. 7 meters”). In exocentric evaluations, estimates were made using environmental elements (e.g., "If the height of this apartment is 3 meters, the 5th floor should be approximately 25 meters"). These estimates are based on the evaluation of visual data.

However, in our study, we aimed to investigate the role of the vestibular system (especially saccular stimuli) in height perception rather than the effects of 3D vision provided by the two eyes, taking necessary precautions to exclude the effect of visual data-based estimates. The results obtained from our study support the assumption that height perception is reliable, as the participants obtained similar results in top-down and bottom-up views.

The background is generally not well written. Many sentences need work and do not entirely make sense. In general the English in the paper is not very good, which is somewhat frustrating when trying to review the work. I don’t think it is my role as a reviewer to correct English. The manuscript should be better proof-read before submission to a journal.

We rewrite the introduction for to clarify the hypothesis and the aims of the study. The manuscript edited by an native speaker after artificial intelligence check.

The hypothesis of the experiment is not well defined. Are you assuming that the lift is stimulating the saccule and that this will somehow affect height perception? Why? Have others shown that (and if so what did they do in their experiments?). It seems quite tenuous. The points about correcting for the height of the subjects are not explained clearly/justified. There is no hypothesis stated, or any rationale given for the extra analysis.

Basically, there are no studies on how our perception of height is formed. In the existing literature, the use of visual data and the role of individuals' cognitive skills in environmental perception are emphasized. However, studies have shown that functional disorders of the saccule have an effect on cognitive skills and can therefore change environmental perception. In this context, in our study, we aimed to evaluate whether there is a perceptual difference between the conditions in which the saccule is stimulated by perilymphatic movement and the conditions in which perilymphatic movement is absent (i.e., not stimulated). Based on your suggestions, edits have been made to the introduction to express this purpose more clearly.

I am not convinced that going 5 floors in a lift is stimulating the saccule to a degree that we would expect a change in height perception (if we are assuming that there is evidence for saccular stimulation affecting height perception – although that is not clear from the manuscript). Also the non-simultaneous measurement of height with the life stimulation means that any affect on height perception could have recovered anyway. I therefore don’t think that the conclusion ‘Mechanical stimulation of the saccule is thought to have a limited role in height perception’ is justified from the work carried out.’

Since there is no previous study in this area, the 5th floor was chosen as the experimental environment. This floor is considered to be both easily accessible and a level where height perception may come to the fore in the visual perception of the participants, rather than floor estimation based on a more horizontal view, such as the first floor. In the first trials, it was observed that the participants calculated the height on the 1st, 2nd, and 3rd floors based on their horizontal viewing angles and the duration of their stay in the elevator. However, it has been found that this does not occur at higher floors. For this reason, the 5th floor was preferred as a level where the existing facilities and fear of heights would be minimal in the participants.

Given these limitations I don’t think that the work is suitable for publication at this time. Possibly the work comparing VR height perception to real-world height perception would be of interest to some in the VR community.

Reviewer’s Comment 1: : Abstract You probably don’t need to put the date.

It might help to say briefly what the mechanical stimulation is in the abstract.

Acoustic immittance (not ‘immitansmetry’)

In the abstract section, date information is omitted and mechanical stimulation is described in detail. Inconsistencies in the terms expressing clinical measurement were identified and the necessary arrangements were made and the expressions were revised to ensure terminological consistency.

Reviewer’s Comment 2: L53 I am not sure about your definition of balance

This definition was made by “Charles M. Plishka” in the text book dated “2015”.

Reviewer’s Comment 3: L54 which image?

Image defines “It refers to the visu

---

## [Decision Letter · Decision Letter 1]

30 May 2025

PONE-D-24-40618R1Comparison of virtual reality and real environment effects on perception of height in healthy individualsPLOS ONE

Dear Dr. Karaman,

Thank you for submitting your manuscript to PLOS ONE. After careful consideration, we feel that it has merit but does not fully meet PLOS ONE’s publication criteria as it currently stands. Therefore, we invite you to submit a revised version of the manuscript that addresses the points raised during the review process.

**ACADEMIC EDITOR: Justify the project and needs more clarity**

We look forward to receiving your revised manuscript.

Kind regards,

Jeyasakthy Saniasiaya, MD, MMed ORLHNS, FEBORLHNS

Academic Editor

PLOS ONE

Additional Editor Comments:

The authors need to focus on justifying the reserach question and the hypothesis clearly

Reviewers' comments:

Reviewer's Responses to Questions

**Comments to the Author**

1. If the authors have adequately addressed your comments raised in a previous round of review and you feel that this manuscript is now acceptable for publication, you may indicate that here to bypass the “Comments to the Author” section, enter your conflict of interest statement in the “Confidential to Editor” section, and submit your "Accept" recommendation.

Reviewer #3: All comments have been addressed

Reviewer #4: All comments have been addressed

2. Is the manuscript technically sound, and do the data support the conclusions?

Reviewer #3: No

Reviewer #4: Yes

3. Has the statistical analysis been performed appropriately and rigorously? 

Reviewer #3: Yes

Reviewer #4: Yes

4. Have the authors made all data underlying the findings in their manuscript fully available?

Reviewer #3: No

Reviewer #4: Yes

5. Is the manuscript presented in an intelligible fashion and written in standard English?

Reviewer #3: Yes

Reviewer #4: Yes

6. Review Comments to the Author

Reviewer #3: The authors have generally tried to relate how saccular function might relate to height perception by using both virtual reality and a mental calculation of the height where participants are positioned. The authors also point out that this exercise would mechanically stimulate saccular function, so that its normal function would have a bearing on height perception.

In my opinion, the claim that these paradigms can stimulate saccule function is not convincing. It is not clear to me how going up or down in a lift could generate linear accelerations that could mechanically stimulate the maculae (an issue that is not rigorously discussed in the text in my opinion). On the other hand, the authors do not describe what kind of results they obtained in the previously performed cVEMP and oVEMP tests (what is considered a normal response by the authors; are there asymmetries in the inter-ear responses of the participants; what was the amplitude and latency of these responses; what was the amplitude and latency of these responses).

I think you cannot extrapolate these conclusions without at least one control experiment with subjects with abnormal macular function (assessed by cVEMPs and oVEMPs), showing that in these cases height perception is significantly altered. There may be other conditions where spatial cognition is impaired (e.g. persistent perceptual postural motion sickness) where cVEMP and oVEMP tests are normal. Also the item discussing the relationship between cognitive dysfunction and vestibular impairment is also weak and needs to be better explored (I suggest Harun, A., Oh, E. S., Bigelow, R. T., Studenski, S., and Agrawal, Y. (2016). Vestibular

impairment in dementia. Otol. Neurotol. 37, 1137-1142. doi: 10.1097/

MAO.0000000000001157; Smith, P. F. (2023). Aging of the vestibular system and its relationship to dementia.

Curr. Opin. Neurol. 37, 83-87. doi: 10.1097/WCO.0000000000001231).

For the above reasons, I consider that this paper is not suitable for publication in PlosOne. I encourage the authors to perform a control experiment that contrasts how subjects with saccular damage might show dysfunction in height perception relative to subjects with normal otolithic assessment.

Reviewer #4: The authors have conducted an interesting experimental study exploring the role of saccular stimulation on height perception in real versus virtual environments. The study is particularly important in an era where virtual reality (VR) is being increasingly adopted and will be used for the foreseeable future.

Comments:

o While the authors have improved on the previous reviewer’s comment and has written their hypothesis in more detail, the hypothesis is still implied rather than explicitly stated and can be improved.

o The manuscript has occasional grammatical errors and awkward phrasing and can be improved there (e.g., Participants were informed in writing with an informed consent form and their signed consent was obtained).

o The authors need to elaborate on the rationale for using 5th floor in their study. The authors can cite prior literature or priori used here.

7. PLOS authors have the option to publish the peer review history of their article (what does this mean?). If published, this will include your full peer review and any attached files.

Reviewer #3: No

Reviewer #4: No

---

## [Author Response · Author response to Decision Letter 2]

11 Jul 2025

July 10th, 2025

Mrs. Jeyasakthy Saniasiaya,

Academic Editor

PLOS ONE

Dear Mrs. Jeyasakthy Saniasiaya,

According to your last email, please find attached a copy of the revised version of our paper, as a possible publication in PLOS ONE:

“COMPARISON OF VIRTUAL REALITY AND REAL ENVIRONMENT EFFECTS ON PERCEPTION OF HEIGHT IN HEALTHY INDIVIDUALS” The comments and suggestions for revision are reflected in the current version of the paper.

A response letter with main changes has been done in this revision together a separated individual response for each reviewer to their comments and a specific reference section to support our answers are attached.

Looking forward to hearing from you.

Thank you.

Yours sincerely,

Response letter to the Reviewers’ Comments on PLOS ONE

Title of the paper: Comparison of virtual reality and real environment effects on perception of height in healthy individuals

First of all, authors would like to thank all the anonymous Reviewers and the Editors again for their efforts and valuable time to review and improve our paper.

Taking into account their constructive suggestions and comments, the paper has been carefully revised following the referees’ comments.

The main changes in the revised manuscript are:

1. Research question and hypothesis

2. cVEMP and saccular functions

3. Spatial cognition

4. Grammar check

We provide below the responses to each referee’s comments.

Additional Editor Comments: The authors need to focus on justifying the reserach question and the hypothesis clearly

Thank you for this valuable suggestion. We have revised the final paragraph of the Introduction to clearly articulate both the rationale and the hypothesis. Specifically, we state: “In this study, we hypothesized that mechanical stimulation of the saccule—induced by vertical acceleration during elevator movement—may influence vertical distance perception in healthy individuals. Specifically, we aimed to investigate whether this effect differs between real and virtual environments.”

Additionally, we elaborated on the physiological basis for this hypothesis and the relevance of investigating this phenomenon in healthy individuals with confirmed normal saccular function.

Reviewer #3

Reviewer’s Comment:

a. In my opinion, the claim that these paradigms can stimulate saccule function is not convincing. It is not clear to me how going up or down in a lift could generate linear accelerations that could mechanically stimulate the maculae (an issue that is not rigorously discussed in the text in my opinion).

We appreciate this important observation. The saccule is known to be responsive to linear vertical accelerations [1], even at low intensities, such as those generated during passive elevator movement. Prior studies have demonstrated modulations in vestibular-evoked myogenic potentials (cVEMP) in response to slow vertical motion or optical illusions simulating elevation [2] While elevator-induced accelerations are indeed subtle compared to active motion tasks, they are within the functional range of saccular detection, particularly for low-threshold mechanoreceptors [1]. Our elevator paradigm was designed to generate these subtle vertical accelerations without visual or auditory cues, thereby engaging the saccule through passive stimulation. This justification has been added and expanded in the Introduction section.

1. Curthoys I. S. (2020). The Anatomical and Physiological Basis of Clinical Tests of Otolith Function. A Tribute to Yoshio Uchino. Frontiers in neurology, 11, 566895. https://doi.org/10.3389/fneur.2020.566895

2. Dündar D. (2022). Investigation of the effect of virtual reality environments on the vestibular system using the VEMP test. Thesis. Istanbul: Medipol University. https://acikerisim.medipol.edu.tr/xmlui/bitstream/handle/20.500.12511/11963/Dundar-Dilara-2022.pdf?sequence=1&isAllowed=y

b. On the other hand, the authors do not describe what kind of results they obtained in the previously performed cVEMP and oVEMP tests (what is considered a normal response by the authors; are there asymmetries in the inter-ear responses of the participants; what was the amplitude and latency of these responses; what was the amplitude and latency of these responses).

Thank you for pointing this out. All participants underwent cVEMP testing to confirm normal saccular function. Only those with clearly reproducible bilateral responses within normative amplitude (typically >100 µV) and latency ranges (p13 latency ~13 ms; n23 latency ~23 ms) were included. Any participant exhibiting absent, delayed, or asymmetrical responses was excluded from the study. A brief summary of these criteria and the screening outcome has been added to the Methods section.

c. I think you cannot extrapolate these conclusions without at least one control experiment with subjects with abnormal macular function (assessed by cVEMPs and oVEMPs), showing that in these cases height perception is significantly altered.

However, this study design was chosen for two main reasons: a. This study was specifically designed as a first step to investigate the perceptual consequences of mechanical and non-mechanical stimulation of the saccule in individuals with normal vestibular function. One of the main objectives was to establish a normative baseline in a population without balance issues and with normal saccule responses (assessed by cVEMP). These baseline data will serve as a reference for future studies involving participants with vestibular dysfunction. b. The examination of the effects of abnormal saccular function responses poses significant challenges due to the compensatory capabilities of the vestibular system and the functional principles of the saccule (unlike the semicircular canals, the saccule is divided into two parts by the striola without a counterpart on the opposite side).

d. There may be other conditions where spatial cognition is impaired (e.g. persistent perceptual postural motion sickness) where cVEMP and oVEMP tests are normal. Also the item discussing the relationship between cognitive dysfunction and vestibular impairment is also weak and needs to be better explored (I suggest

Harun, A., Oh, E. S., Bigelow, R. T., Studenski, S., and Agrawal, Y. (2016). Vestibular impairment in dementia. Otol. Neurotol. 37, 1137-1142. doi: 10.1097/ MAO.0000000000001157;

Smith, P. F. (2023). Aging of the vestibular system and its relationship to dementia. Curr. Opin. Neurol. 37, 83-87. doi: 10.1097/WCO.0000000000001231).

Thank you for this constructive suggestion. We have revised the Introduction to incorporate recent findings on the vestibular–cognition relationship. In particular, we now reference Harun et al. (2016), who reported vestibular deficits in patients with dementia, and Smith (2023), who highlighted age-related vestibular degeneration as a contributing factor to cognitive decline. These additions help clarify that vestibular inputs, including those from the otolith organs, are integrally involved in spatial awareness and cognitive functioning.

Reviewer #4

Reviewer’s Comment:

a. While the authors have improved on the previous reviewer’s comment and has written their hypothesis in more detail, the hypothesis is still implied rather than explicitly stated and can be improved.

In this study, we hypothesized that mechanical stimulation of the saccule—induced by vertical acceleration during elevator movement—may influence vertical distance perception in healthy individuals. Specifically, we aimed to investigate whether this effect differs between real and virtual environments

b. The manuscript has occasional grammatical errors and awkward phrasing and can be improved there (e.g., Participants were informed in writing with an informed consent form and their signed consent was obtained).

We have carefully revised the manuscript for grammatical correctness and clarity. The sentence mentioned by the reviewer has been corrected to:

“All participants were provided with written informed consent forms, and written consent was obtained prior to participation.”

We also reviewed the entire manuscript for similar issues and made necessary corrections to improve readability and scientific tone.

c. The authors need to elaborate on the rationale for using 5th floor in their study. The authors can cite prior literature or priori used here.

Thank you for pointing this out. The fifth floor (~24 meters above ground) was selected based on pilot testing that indicated participants could not use exocentric cues (e.g., counting floors) to estimate height in either the real or virtual environment. This level of elevation was also sufficient to engage height perception without inducing discomfort or height phobia. The simulation environment was specifically designed to eliminate such cues and provide a perceptually neutral platform. We have included this rationale in the revised Methods and Introduction sections, and where applicable, we cited relevant virtual height perception studies.

---

## [Decision Letter · Decision Letter 2]

1 Aug 2025

Comparison of virtual reality and real environment effects on perception of height in healthy individuals

PONE-D-24-40618R2

Dear Dr. Karaman,

We’re pleased to inform you that your manuscript has been judged scientifically suitable for publication and will be formally accepted for publication once it meets all outstanding technical requirements.

Kind regards,

Jeyasakthy Saniasiaya, MD, MMed ORLHNS, FEBORLHNS

Academic Editor

PLOS ONE

Additional Editor Comments (optional):

Authors have adequately revised

Reviewers' comments:

Reviewer's Responses to Questions

**Comments to the Author**

1. If the authors have adequately addressed your comments raised in a previous round of review and you feel that this manuscript is now acceptable for publication, you may indicate that here to bypass the “Comments to the Author” section, enter your conflict of interest statement in the “Confidential to Editor” section, and submit your "Accept" recommendation.

Reviewer #4: All comments have been addressed

2. Is the manuscript technically sound, and do the data support the conclusions?

Reviewer #4: Yes

3. Has the statistical analysis been performed appropriately and rigorously? 

Reviewer #4: Yes

4. Have the authors made all data underlying the findings in their manuscript fully available?

Reviewer #4: Yes

5. Is the manuscript presented in an intelligible fashion and written in standard English?

Reviewer #4: Yes

6. Review Comments to the Author

Reviewer #4: The revised manuscript presents a novel and thoughtfully designed study comparing perception of height between virtual and real environments, with a focus on the influence of vertical acceleration and saccular function. The authors have addressed previous concerns comprehensively.

1.The revised Introduction now clearly explains the physiological rationale for exploring saccular stimulation via elevator-induced vertical acceleration. The hypothesis is also more explicitly stated. Thank you for this revision.

2.The rationale for selecting the fifth floor as the experimental height is reasonable and now well-supported by the discussion of pilot testing and absence of exocentric cues. While no direct prior literature is cited for this specific elevation, the explanation is acceptable, especially given the virtual and perceptual design focus.

3.Consider consistently using either "vertical distance perception" or "height perception" throughout the paper. While they are closely related, consistent terminology may aid clarity.

7. PLOS authors have the option to publish the peer review history of their article (what does this mean?). If published, this will include your full peer review and any attached files.

Reviewer #4: No

---

## [Editor Report · Acceptance letter]

PONE-D-24-40618R2

PLOS ONE

Dear Dr. Karaman,

I'm pleased to inform you that your manuscript has been deemed suitable for publication in PLOS ONE. Congratulations! Your manuscript is now being handed over to our production team.

Kind regards,

on behalf of

Dr. Jeyasakthy Saniasiaya

Academic Editor

PLOS ONE